# Sleep Quality and Its Associations with Physical and Mental Health-Related Quality of Life among University Students: A Cross-Sectional Study

**DOI:** 10.3390/ijerph19052874

**Published:** 2022-03-01

**Authors:** Matteo Carpi, Claudia Cianfarani, Annarita Vestri

**Affiliations:** 1Department of Psychology, Sapienza University of Rome, 00185 Rome, Italy; 2Department of Public Health and Infectious Diseases, Sapienza University of Rome, 00185 Rome, Italy; dottoressa.cianfarani@gmail.com (C.C.); annarita.vestri@uniroma1.it (A.V.)

**Keywords:** sleep quality, insomnia, health-related quality of life, perceived stress, university students, COVID-19 impact

## Abstract

The association between sleep problems and quality of life has been well documented and the COVID-19 pandemic seemingly had an impact on both sleep quality and health-related quality of life (HRQoL). However, recent evidence about this relationship among university students is limited. The aims of this study are to investigate the prevalence of poor sleep quality and insomnia and to explore the associations between these outcomes, perceived stress, and HRQoL among Italian university students. An anonymous questionnaire comprising the Pittsburgh Sleep Quality Index, the Insomnia Severity Index, the Short Form-12 health survey, and the Perceived Stress Scale was administered to a convenience sample of 1279 students (1119 females and 160 males, mean age: 23.4 ± 2.5 years) attending one of the largest Italian universities. A total of 65% of the participants showed poor sleep quality, whereas 55% reported insomnia symptoms. Students reporting poor sleep quality and insomnia obtained higher perceived stress scores and lower physical and mental HRQoL scores. Controlling for health-related variables and perceived stress, hierarchical regression analyses showed that sleep quality components added a significant contribution to the prediction of both physical (ΔR^2^ = 0.1) and mental (ΔR^2^ = 0.02) HRQoL. As a whole, these findings confirm the relevance of sleep for university students’ well-being and might inform the development of health promotion interventions for this population.

## 1. Introduction

Poor sleep quality, reduced sleep duration, and sleep disorders, such as insomnia and obstructive sleep apnea, have been widely shown to be associated with adverse outcomes for both physical and mental health, including cardiovascular and metabolic diseases [1,2,3,4], anxiety and depression [5,6,7], and reduced cognitive functioning [8], consistently with the acknowledged role of sleep in the regulation of biological processes pivotal for health [9].

Sleep quality has been conceptualized as a construct comprising both quantitative-objective and subjective aspects of sleep [10,11]. Sleep disorders are highly prevalent among young adults [12] and poor sleep quality seems to be particularly common among university students, with large studies conducted across different socio-cultural contexts [13,14,15,16,17] reporting prevalence rates between 50% and 70% for poor sleep quality assessed with the Pittsburgh Sleep Quality Index [18]. In this population, significant associations have been reported between poor sleep, mental health problems [19], and academic performance [20]. Given the well-established bidirectional nature of the relationship between poor sleep and distress [5,6,21,22], improving sleep quality should be a primary objective of preventive mental health interventions for young adults.

Quality of life is a broad concept that refers to the subjective perception of one’s well-being and position in life [23], and health-related quality of life (HRQoL) pertains to the health aspects of quality of life, which directly reflect on physical and mental well-being. Given the above-mentioned role of sleep in health and its relations with both mental and physical conditions whose relevance for well-being is evident, it is not surprising that the association between sleep quality and quality of life has been widely investigated both in clinical [24,25,26] and community samples [27,28,29,30]. For example, in a large community sample of young adults from the U.S., Chen, Gelaye, and Williams [27] found a significant association between sleep disturbances and a low health-related quality of life, and Marques and colleagues [29] showed that subjective sleep quality was related to the main aspects of quality of life, including physical and psychological health. Similarly, Clement-Carbonell et al. [28] reported significant associations between sleep quality and both physical and mental health-related quality of life in a sample of Spanish university students, with the latter association being stronger than the former.

The recent outbreak of the COVID-19 pandemic seemingly had a negative impact on both sleep quality [31] and health-related quality of life of the general population [32,33]. Several studies conducted in Italy during the first lockdown period in 2020 reported increased sleep difficulties in association with high levels of depression, anxiety, and stress on samples mostly comprising university students [34,35,36,37]. However, to our knowledge, no results have been reported to date with respect to the relationship between sleep quality and health-related quality of life in Italy, neither before nor after the occurrence of the COVID-19 pandemic. Based on these premises, this study aims to investigate sleep quality and insomnia symptoms among Italian university students in the late phase of the COVID-19 pandemic, to examine the relationships between these outcomes, perceived stress, and health-related quality of life, and to explore the associations between specific aspects of sleep quality and physical and mental health-related quality of life controlling for the influence of socio-demographic and health-related variables and psychological distress.

## 2. Materials and Methods

### 2.1. Participants

An online cross-sectional survey was conducted from March 2021 to June 2021. An invitation to participate was delivered through mailing lists and university social network groups to students enrolled at the Sapienza University of Rome. A total of 1288 students attending undergraduate or postgraduate taught courses from diverse departments and faculties answered the survey. Consistently with the purpose of investigating the condition of young adults university students, subjects older than 35 years (n = 9) were excluded. No further exclusion criteria were applied.

All participants were informed about the aims of the study, took part in the survey voluntarily with confidentiality and anonymity assurances, and provided informed consent. The study protocol was approved by the competent Ethics Committee at Sapienza University of Rome (protocol number 0308/2021).

### 2.2. Measures

The administered survey consisted of an introductory section gathering socio-demographic and health-related information, followed by the Italian versions of the Pittsburgh Sleep Quality Index (PSQI) [18], the Insomnia Severity Index (ISI) [38], the 10-item Perceived Stress Scale (PSS-10) [39,40], and the Short Form-12 questionnaire (SF-12) [41]. Cronbach’s alpha coefficients (α) were calculated in order to examine the reliability of the scales with respect to the conventional criteria [42].

Sleep quality was assessed with the Italian version of the PSQI [43]. This well-known questionnaire is composed of 19 items with different response formats (open-ended or 5-point Likert scales) that investigate typical sleep duration, sleep time habits, and sleep-related disturbances, and measures 7 dimensions (subjective sleep quality, sleep latency, sleep duration, habitual sleep efficiency, sleep disturbances, use of sleep medications, and daytime dysfunction) with aggregated scores ranging from 0 to 3. The total score (range: 0–21) was obtained by summing the dimensions’ scores, with higher scores corresponding to a worse sleep quality. A cut-off of 5 for the global score was identified for poor sleep quality [18]. An alternative scoring model was proposed for the PSQI with three factors [44], namely perceived sleep quality (given by the sum of the scores of subjective sleep quality, sleep latency, and use of sleep medications), sleep efficiency (given by the sum of the scores of sleep duration and habitual sleep efficiency), and daily disturbances (given by the sum of the scores of sleep disturbances and daytime dysfunction). In order to obtain a greater amount of information, both scoring models (global score and three factors) were used in this study. In the original validation study [43], the Italian PSQI successfully discriminated between healthy controls and patients with various sleep complaints and showed a high reliability (Cronbach’s α = 0.83). Internal consistency for the seven dimensions’ scores was acceptable in the sample examined in this study with α = 0.7. Information about specific sleep variables, such as sleep duration and sleep latency, was derived from the PSQI items.

Symptoms of insomnia were measured with the ISI, a seven-item questionnaire assessing subjective sleep difficulties, which has been shown to be reliable (Cronbach’s α = 0.74) and sensitive to detect changes in perceived insomnia severity after treatment [38]. Items are rated on a 5-point response scale and their sum provides a global score ranging from 0 to 28. A total of 4 severity categories were identified for the total score: no insomnia (score range: 0–7), subthreshold insomnia (score range: 8–14), moderate insomnia (score range: 15–21), and severe insomnia (score range: 22–28). The ISI showed acceptable internal consistency in this study with α = 0.84.

Perceived stress was measured with the 10-item version of the PSS, a widely used brief scale. Items are both negatively (e.g., ‘In the last month, how often have you been upset because of something that happened unexpectedly?’) and positively phrased (e.g., ‘In the last month, how often have you felt confident about your ability to handle your personal problems?’), refer to the last month, and are rated on a five-point Likert scale. The total score ranges between 0 and 40, with higher scores indicating a higher level of perceived stress, and associations were reported between perceived stress scores and symptoms of depression and anxiety [45]. The Italian version of the scale showed good psychometric properties in terms of reliability and concurrent validity (Cronbach’s α = 0.74 and significant correlation with depressive symptoms), even in comparison with the original 14-item version of the instrument [46]. Internal consistency was adequate in this study with α = 0.81.

The SF-12 was used to measure the health-related quality of life. The questionnaire consisted of 12 items with different response formats and evaluated 8 dimensions (namely, physical activity, role and physical health, role and emotional state, mental health, physical pain, general health, vitality, and social activities). These dimensions are aggregated in two scores, the physical component summary (PCS) and the mental component summary (MCS), which reflect, respectively, the physical and mental health-related quality of life and showed good empirical validity and acceptable reliability in repeated administrations (test–retest reliability coefficients were 0.89 for the PCS and 0.76 in the original validation study [41]). Scores for both components are standardized (M = 50, SD = 10), with higher scores corresponding to a better quality of life. The Italian adaptation of the questionnaire [47] was used in this study and satisfactory reliability was found for the global score (α = 0.81).

### 2.3. Statistical Analyses

Descriptive and inferential statistical analyses were conducted using IBM SPSS software (version 25.0, IBM Corp., Armonk, NY, USA). Continuous variables were summarized by the means and standard deviations or medians and interquartile ranges; the categorical variables were summarized by counts and percentages with 95% confidence intervals. Questionnaires’ scores were considered as continuous variables.

The association between categorical variables was examined with a chi-squared test, whereas the Student’s *t*-test or Mann–Whitney U test were used to compare continuous variables according to the data distribution, and Cohen’s d was calculated as a measure of effect size, considering d = 0.20 as a small effect size, d = 0.50 as a medium effect size, and d = 0.80 as a large effect size [48]. Pearson’s correlation coefficients were calculated to evaluate the bivariate relationships among the scores of the questionnaires. Hierarchical multiple regression analyses were conducted to explore the association between the investigated variables and health-related quality of life. For all the analyses performed, *p*-values below 0.05 were considered statistically significant.

## 3. Results

### 3.1. Participants’ Characteristics and Sleep Status

The final sample comprised 1279 university students, with a large prevalence of females (1119 females and 160 males, mean age: 23.4 ± 2.5 years). Participants’ characteristics and questionnaires’ scores are reported in Table 1. The mean sleep duration was 7.0 ± 1.1 h and 32.6% (n = 417; 95% CI: 30–35.2%) of the students reported less than 7 h of sleep per night. With respect to the sleep quality and insomnia severity, 64.5% (n = 825; 95% CI: 61.9–67.1%) of the sample obtained a score above 5 on the PSQI, indicating poor sleep quality, and 54.8% (n = 701; 95% CI: 52.1–57.5%) scored above the cut-off of 7 on the ISI, with 41.2% of the participants (n = 527; 95% CI: 38.5–44%) reporting subthreshold insomnia symptoms, 12.4% (n = 159; 95% CI: 10.7–14.4%) reporting moderate insomnia symptoms, and 1.2% (n = 15; 95% CI: 0.68–2%) reporting severe insomnia symptoms. No significant association was found between gender (female/male) and both sleep quality (good/poor; χ^2^ = 1.2, *p* = 0.27) and insomnia severity (below cut-off/above cut-off; χ^2^ = 0.64, *p* = 0.43), although higher mean scores were found for females on the PSQI (t = 3.04, *p* < 0.01, d = 0.26).

### 3.2. Differences in Perceived Stress and HRQoL According to Sleep Quality and Insomnia Severity

The mean PSS-10, PCS, and MCS scores grouped according to sleep quality (good sleepers vs. poor sleepers) and insomnia severity (below cut-off insomnia symptoms vs. above cut-off insomnia symptoms) are reported in Table 2. Poor sleep quality was associated with higher distress and a worse physical and mental health-related quality of life, with poor sleepers showing higher PSS-10 scores (t = −10.7, *p* < 0.001, d = 0.62) and lower PCS (t = 9.0, *p* < 0.001, d = 0.53) and MCS scores (t = 10.7, *p* < 0.001, d = 0.63) with consistently medium-effect sizes. The same pattern with similar-effect sizes was observed for subjects reporting relevant insomnia symptoms (respectively for PSS-10, PCS, and MCS: t = −12.0, *p* < 0.001, d = 0.66; t = 11.2, *p* < 0.001, d = 0.64; t = 10.6, *p* < 0.001, d = 0.59).

### 3.3. Relationship between Sleep Quality and HRQoL

The correlation coefficients between the PSQI, the ISI, the PSS-10, and the SF-12 components scores are reported in Table 3. All the examined relations were statistically significant except for the correlation between the PCS and the MCS (r = −0.037, *p* = 0.19). Consistently with the similar nature of the measured constructs, significant and large correlations were found between the PSQI score and the ISI score (r = 0.726, *p* < 0.001), and between the PSS-10 score and the MCS score (r = −0.65, *p* < 0.001). In addition, perceived stress, physical health-related quality of life, and mental health-related quality of life all showed significant low–medium correlations with both sleep quality (respectively r = 0.393, *p* < 0.001; r = −0.350, *p* < 0.001; r = −0.353, *p* < 0.001) and insomnia severity (respectively r = 0.382, *p* < 0.001; r = −0.323, *p* < 0.001; r = −0.339, *p* < 0.001).

In order to examine the association between sleep quality and health-related quality of life controlling for the impact of demographic and health-related factors and perceived stress better, the results of bivariate analyses were further explored by conducting two hierarchical multiple regressions with the PCS and the MCS scores as the dependent variables. For both models, the same independent variables were entered in two subsequent steps: gender (with female as baseline), age, smoking habit (with 0 = non-smoker and 1 = smoker), physical exercise (with 0 = exercise less than 2 times a week and 1 = exercise 2 or more times a week), body mass index (BMI; in kg/m^2^ as a continuous variable), and the PSS-10 score in Step 1, and the three dimensions of the PSQI identified by Cole et al. (2006) [44], namely sleep efficiency, perceived sleep quality, and daytime disturbances in Step 2. The sub-dimensions were preferred over the PSQI global score in order to examine in detail the associations among the specific aspects of sleep quality and health-related quality of life. 

Variance inflation factors (VIFs) were below 2 and the Durbin–Watson statistic values were around 2 for all the independent variables, and a visual inspection revealed no significant asymmetry in residuals distributions for both models, thus relevant multicollinearity was excluded. The coefficients and summary statistics for the two models predicting the PCS and the MCS scores are reported in Table 4 and Table 5.

In the first model (Step 2 in Table 4; R^2^ = 0.17, F = 27.81, *p* < 0.001), a statistically significant association with physical health-related quality of life was found for physical exercise (*p* < 0.01, β = 0.09), body mass index (*p* < 0.05, β = −0.05), perceived sleep quality (*p* < 0.01, β = −0.09), and daily disturbances (*p* < 0.001, β = −0.29), but not for sleep efficiency (*p* = 0.11). The addition of sleep quality dimensions in Step 2 significantly improved the model R^2^ (ΔR^2^ = 0.1, F = 50.33, *p* < 0.001).

On the other hand, in the model predicting the MCS scores (Step 2 in Table 5; R^2^ = 0.44, F = 111.46, *p* < 0.001), mental health-related quality of life was found to be associated with perceived stress (*p* < 0.001, β = −0.59) and daily disturbances (*p* < 0.001, β = −0.12), whereas no associations were found with perceived sleep quality (*p* = 0.06) and sleep efficiency (*p* = 0.81). Again, the contribution of the sleep quality dimensions block was significant (ΔR^2^ = 0.02; F = 12.8; *p* < 0.001).

## 4. Discussion

This study examined sleep quality, insomnia symptoms, perceived stress, and health-related quality of life in a wide convenience sample of university students attending one of the largest Italian universities, in the late phase of the COVID-19 pandemic, with the aim to assess the prevalence of sleep problems and to explore the relationships between specific dimensions of sleep quality and both physical and mental health-related quality of life. Descriptive statistics showed that up to 65% of the sample reported poor sleep quality according to the PSQI, and 33% had a habitual sleep duration of less than 7 h, which is the minimum amount of sleep conventionally recommended for young adults and adults [49]. Moreover, 55% of the participants reported relevant symptoms of insomnia on the ISI, with approximately 14% of the sample obtaining a score indicative of clinical insomnia. Although a major proneness to sleep problems was previously reported for females [50,51], no substantial associations were found between gender and poor sleep or insomnia in our study, albeit females obtained a higher mean score on the PSQI corresponding to worse sleep quality. These results are fundamentally in line with those reported on similar samples both before [13,14,15,16,17] and after the COVID-19 pandemic outbreak [34,35,37], with previous estimates for poor sleep quality ranging between 50% and 70%, and further confirm that sleep problems represent a relevant issue among university students. Furthermore, high levels of perceived stress and low mental health-related quality of life were observed in the examined sample. The PSS mean score is consistent with that reported for a large general population Italian sample during the first COVID-19 lockdown [52] and significantly higher than those found in samples of university students in the same period [53,54], whereas the mean standard score observed for the SF-12 index MCS is inferior in comparison with that reported in a similar sample during the COVID-19 pandemic [55] and almost two standard deviations lower than the mean score for the Italian normative population [47]. Since restrictive measures with several periods of lockdown have been adopted in Italy between 2020 and 2021, such alarming results are likely to be attributable at least in part to the well-documented impact of the pandemic and the related policies on mental health [56], with seemingly worse outcomes reported for young people [57,58].

Higher levels of perceived stress and worse physical and mental health-related quality of life were found for participants with poor sleep quality and mild insomnia symptoms. Effect sizes were similar for sleep quality and insomnia, in line with the results of correlation analyses, which showed a strong relationship between the PSQI and the ISI consistently with previous findings [59]. Furthermore, again consistently with the aforementioned literature concerning the relationships between sleep and mental and physical health, a noticeable overlap was found between sleep variables (namely, sleep quality and insomnia symptoms) and perceived stress, mental, and physical HRQoL.

Regression analyses further explored these associations, showing that sleep quality components of sleep efficiency, perceived sleep quality, and daily disturbances (i.e., common sleep problems, such as difficulties with falling asleep, awakenings, bad dreams, snoring, and their daily consequences), taken together, uniquely explained 10% of the variance in physical HRQoL and 2% of the variance in mental HRQoL controlling for relevant demographic and health-related variables and perceived stress levels. Thus, while the impact of sleep quality on the physical dimension of HRQoL appears rather consistent, its impact on the mental dimension seems to be less significant, but this last result should be interpreted while taking into consideration the large overlap with variance explained by the perceived stress in the model predicting MCS scores. Indeed, with respect to single sleep quality components, perceived sleep quality and daily disturbances were found to be significant predictors of physical HRQoL, with worse sleep quality and higher levels of disturbances corresponding to lower PCS scores, and daily disturbances were also associated with lower MCS scores, even controlling for the impact of perceived stress, which was previously identified as a robust predictor of mental HRQoL [60]. These findings are substantially consistent with those reported in previous studies that considered the same variables and used the same measure of HRQoL (namely, the SF-12), although some differences should be considered. In particular, Darchia and colleagues [24] found significant associations between sleep quality (PSQI global score) and both physical and mental HRQoL after controlling for socio-demographic variables, BMI, and depression in a sample of 395 subjects aged between 20 and 60 years from Georgia; however, in their study, the magnitude of the unique association between sleep quality and physical HRQoL was less consistent than that we observed in our sample. However, this discrepancy might be attributable to the wider age range of the subjects included in the research of Darchia et al., since it is likely that several other uncontrolled factors beyond sleep quality had an impact on the older subjects’ perceptions of their physical health. On the other hand, in the study conducted by Clement-Carbonell and colleagues [28], the same pattern of associations was observed among 337 university students from Spain after controlling for stress and health habits (diet and exercise). Consistently with our findings, significant relations were found between physical HRQoL and both the subjective sleep quality and the sleep disturbances components of the PSQI; however, in comparison with the result of our study, the association observed between sleep quality components and mental HRQoL is stronger. This disparity might be due to the above-mentioned variance overlap between the PSS-10 and the SF-12 MCS, and to the overall high level of perceived stress and low mental HRQoL we found in our sample.

Ultimately, our results show a high prevalence of poor sleep among Italian university students during the COVID-19 pandemic and support the relevance of sleep quality for young people’s overall quality of life, highlighting the importance of the development of effective preventive interventions and treatment strategies for sleep problems tailored to this population. Indeed, the effectiveness of cognitive behavioral therapy for insomnia (CBT-I) with respect to sleep outcomes and daytime functioning has been widely proved both in the general population and among university students, and also in comparison with other treatments [61,62,63,64,65,66]. In particular, among the many outcomes considered, CBT-I showed high effectiveness in reducing insomnia symptoms and ameliorating sleep quality measured with the same instruments employed in this study (namely, the ISI and the PSQI) [66], which encompass both specific sleep-related problems and daytime dysfunctions. The treatment consists of several components (i.e., sleep-focused psychoeducation, behavioral strategies, and cognitive therapy targeting dysfunctional beliefs about sleep), and its modular structure makes it a feasible solution. Furthermore, CBT-I has been successfully adapted to be delivered as a self-help intervention or in a digital format [67,68], and thus might easily be integrated into preventive and recovery interventions targeting stress management and other mental health issues. In these contexts, dedicated sleep interventions might represent a convenient and strategic resource because of their positive effects on factors whose associations with both physical functioning and mental well-being have been explored in this study. In addition, they are likely to have a positive impact on emotional symptoms [69] and to enhance the effect of other mental health interventions capitalizing on the recursive relationship between sleep and psychological well-being, ultimately improving quality of life.

The major strength of this study is that it is the first to directly examine the relationship between sleep quality and health-related quality of life after the outbreak of the COVID-19 pandemic in a relevant sample of university students. However, the study had several limitations, which should be critically considered. First, the survey was delivered to a convenience sample of students from a single university, which was not representative of the condition of the Italian university student population. A large majority of the participants were female (87.5%) and, given the previously reported associations between gender and the investigated variable [51,70,71], this highly imbalanced distribution might have had an influence on the results of the analyses conducted in this study. Indeed, the high prevalence of female students in our sample might have inflated the regression coefficients and our results might not be relevant with respect to the male students who were scarcely represented in this study. Accordingly, the results of the between-gender comparisons conducted should be considered with caution. Moreover, the online recruitment procedure made it impossible to control the quality of the responses and a reliable response rate could not be obtained. Thus, selection bias could have occurred and sample self-selection should indeed be considered as a possible competitive explanation for the high rates of sleep problems and distress observed. In fact, since the participants chose to take part in the study after receiving information about its aims and objectives, it is possible that students more sensitive to the investigated domains (i.e., those suffering from sleep problems) were more inclined to participate, and such a selection process might have produced over-estimated prevalence rates for both poor sleep quality and insomnia. For all these reasons, the generalizability of the presented findings could be limited and a comparison with previous results should be conducted cautiously. Furthermore, our study design was cross-sectional and causal inferences cannot be drawn from the results; directional paths of association should be tested in longitudinal studies, which could help to disentangle and comprehend the characteristic bidirectional interaction between sleep and health-related quality of life better. With respect to the adopted methodology, we exclusively used self-report standardized questionnaires in order to reach a wide sample size easily, but more in-depth information about sleep habits could have been obtained with an extensive multi-method assessment using idiographic sleep diaries and objective measures, such as actigraphy. Finally, relevant data concerning other sleep disorders (e.g., obstructive sleep apnea, circadian rhythm disorders, and nightmares); health-relevant behaviors, such as substance misuse and the use of electronic devices; and specific psychological variables, such as resilience and coping, have not been gathered in our study, despite their role in the relationship between sleep and well-being, which is likely to be significant [24,72,73,74]. Additionally, no information was collected about contextual and individual factors related to the COVID-19 pandemic (e.g., exposure to contagion/infection, fear of contagion, and feelings of loneliness and isolation), and thus precise inferences about the possible impact of the subjective experience of the pandemic situation on the reported findings cannot be made.

## 5. Conclusions

In conclusion, our study showed high rates of poor sleep quality and insomnia in a sample of students attending a large Italian university, and highlighted a significant association between the major components of sleep quality and physical and mental health-related quality of life after controlling for the effect of health-relevant factors and perceived stress. These findings can inform the design of treatments and health-promotion interventions. Nonetheless, further studies should address these relationships to confirm their relevance and to explore the role of other possible mediating and moderating variables.

## Figures and Tables

**Table 1 ijerph-19-02874-t001:** Participants’ (N = 1279) demographic and health-related characteristics, sleep problems, and mean scores in the PSQI, the ISI, the PSS-10, and the SF-12 questionnaires.

Variable	N (%)	Mean (SD)
Gender		
Female	1119 (87.5)	
Male	160 (12.5)	
Age		23.4 (2.5)
Smoking habit		
Non-smoker	670 (52.4)	
Regular smoker	413 (32.3)	
Occasional smoker	196 (15.3)	
Physical exercise		
≥2 times per week	487 (38.1)	
<2 times per week	792 (61.9)	
BMI (kg/m^2^)		
<18.5 (underweight)	144 (11.3)	
18.5 to 24.99 (normal weight)	929 (72.6)	
≥25 (overweight)	206 (16.1)	
Sleep duration (hours)		7.0 (1.1)
Sleep quality (PSQI)		8.7 (5.2)
Good (PSQI score ≤ 5)	454 (35.5)	
Poor (PSQI score > 5)	825 (64.5)	
Insomnia severity (ISI)		7.1 (3.2)
No insomnia (ISI score ≤ 7)	578 (45.2)	
Subthreshold insomnia (ISI score 8–14)	527 (41.2)	
Moderate insomnia (ISI score 15–21)	159 (12.4)	
Severe insomnia (ISI score > 21)	15 (1.2)	
PSS-10		25.1 (6.4)
SF-12		
PCS—physical component summary		52.6 (7.4)
MCS—mental component summary		33.5 (9.9)

BMI: Body Mass Index; PSQI: Pittsburgh Sleep Quality Index; ISI: Insomnia Severity Index; PSS-10: 10-item Perceived Stress Scale; and SF-12: Short Form-12.

**Table 2 ijerph-19-02874-t002:** Differences in perceived stress, physical health-related quality of life, and mental health-related quality of life scores according to sleep quality and the presence of insomnia.

	Sleep Quality (PSQI)				Insomnia (ISI)			
	Good	Poor			No Insomnia	Insomnia		
	Mean (SD)	Mean (SD)	t	d	Mean (SD)	Mean (SD)	t	d
PSS-10	22.6 (6.5)	26.4 (5.9)	−10.7 ***	0.62	22.9 (6.3)	26.9 (5.8)	−12.0 ***	0.66
PCS	55.1 (6.0)	51.3 (7.7)	9.0 ***	0.53	55.1 (5.9)	50.6 (7.8)	11.2 ***	0.64
MCS	37.4 (10.3)	31.4 (9.2)	10.7 ***	0.63	36.6 (10.5)	30.9 (8.8)	10.6 ***	0.59

PSQI: Pittsburgh Sleep Quality Index; ISI: Insomnia Severity Index; PSS-10: 10-item Perceived Stress Scale; PCS: Short Form–12 Physical Component Summary; and MCS: Short Form–12 Mental Component Summary. *** *p* < 0.001.

**Table 3 ijerph-19-02874-t003:** Pearson’s correlation coefficients between sleep quality (PSQI), insomnia severity (ISI), perceived stress (PSS-10), physical health-related quality of life (SF-12 PCS), and mental health-related quality of life (SF-12 MCS) scores.

	1.	2.	3.	4.	5.
1. PSQI	1				
2. ISI	0.726 ***	1			
3. PSS-10	0.382 ***	0.393 ***	1		
4. PCS	−0.323 ***	−0.350 ***	−0.196 ***	1	
5. MCS	−0.339 ***	−0.353 ***	−0.650 ***	−0.037	1

PSQI: Pittsburgh Sleep Quality Index; ISI: Insomnia Severity Index; PSS: 10-item Perceived Stress Scale; PCS: Short Form–12 Physical Component Summary; and MCS: Short Form–12 Mental Component Summary. *** *p* < 0.001.

**Table 4 ijerph-19-02874-t004:** Hierarchical multiple regression model with physical health-related quality of life (SF-12 PCS) as the dependent variable.

	Step 1			Step 2		
	Coefficient (SE)	t	Beta	Coefficient (SE)	t	Beta
Variables						
Step 1						
Gender	−0.29 (0.62)	−0.47	−0.01	−0.58 (0.59)	−0.98	−0.03
Age	−0.15 (0.08)	−1.84	−0.05	−0.14 (0.08)	−1.83	−0.05
Smoking habit	−1.6 (0.4)	−3.95 ***	−0.11	−1.21 (0.38)	−3.14	−0.08
Physical exercise	1.31 (0.42)	3.15 **	0.09	1.31 (0.39)	3.33 **	0.09
BMI (kg/m^2^)	−0.13 (0.06)	−2.25 *	−0.06	−0.11 (0.05)	−2.02 *	−0.05
PSS-10	−0.2 (0.03)	−6.25 ***	−0.17	−0.02 (0.03)	−0.69	−0.02
Step 2						
Sleep efficiency				−0.26 (0.16)	−1.6	−0.05
Perceived sleep quality				−0.35 (0.13)	−2.76 **	−0.09
Daily disturbances				−2.16 (0.22)	−9.73 ***	−0.29
Summary statistics						
Model F	14.82 ***			27.81 ***		
R^2^	0.07			0.17		
Adjusted R^2^	0.06			0.16		
R^2^ change F	14.82 ***			50.33 ***		

BMI: Body Mass Index; PSQI: Pittsburgh Sleep Quality Index; PSS-10: 10-item Perceived Stress Scale; PCS: Short Form–12 Physical Component Summary; and MCS: Short Form–12 Mental Component Summary. Beta: standardized regression coefficient. * *p* < 0.05; ** *p* < 0.01; *** *p* < 0.001.

**Table 5 ijerph-19-02874-t005:** Hierarchical multiple regression model with mental health-related quality of life (SF-12 MCS) as the dependent variable.

	Step 1			Step 2		
	Coefficient (SE)	t	Beta	Coefficient (SE)	t	Beta
Variables						
Step 1						
Gender	−0.86 (0.66)	−1.3	−0.03	−0.99 (0.65)	−1.53	−0.03
Age	−0.07 (0.09)	−0.82	−0.02	−0.08 (0.09)	−0.87	−0.02
Smoking habit	0.17 (0.43)	0.4	0.01	0.39 (0.43)	0.9	0.02
Physical exercise	0.55 (0.44)	1.25	0.03	0.55 (0.44)	1.25	0.03
BMI (Kg/m^2^)	0.04 (0.06)	0.65	0.01	0.05 (0.06)	0.86	0.02
PSS-10	−1.03 (0.03)	−29.91 ***	−0.65	−0.93 (0.04)	−24.71 ***	−0.59
Step 2						
Sleep efficiency				0.04 (0.18)	0.24	0.01
Perceived sleep quality				−0.26 (0.14)	−1.86	−0.05
Daily disturbances				−1.2 (0.25)	−4.92 ***	−0.12
Summary statistics						
Model F	156.4 ***			111.46 ***		
R^2^	0.43			0.44		
Adjusted R^2^	0.42			0.44		
R^2^ change F	156.4 ***			12.82 ***		

BMI: Body Mass Index; PSQI: Pittsburgh Sleep Quality Index; PSS-10: 10-item Perceived Stress Scale; PCS: Short Form–12 Physical Component Summary; and MCS: Short Form–12 Mental Component Summary. Beta: standardized regression coefficient. *** *p* < 0.001.

## Data Availability

The data that support the results of this study are available from the corresponding author upon reasonable request.

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
