# Peer review of "Sleep Quality and Its Associations with Physical and Mental Health-Related Quality of Life among University Students: A Cross-Sectional Study"

_ijerph, 2022, doi:10.3390/ijerph19052874_

Round 1

Reviewer 1 Report

This study evaluated the association between sleep quality and physical and mental health-related quality of life by hierarchical regression analysis.

  1. The first sentence of the abstract is hard to follow.

  1. Line 30, obstructive sleep apnea is more commonly used.

  1. Line 150-Line 152, is it a part of this manuscript?

  1. 1119 females vs. 160 males. The gender distribution is highly imbalanced. Did you use any method to handle it in the regression analysis? Does this affect the results?

  1. You mentioned the COVID-19 pandemic. Is poor sleep quality related to the COVID-19 pandemic in this study?

  1. What method did you use to verify the reliability of questionnaires responses?

  1. What are your eligibility criteria for this study?

Author Response

Thank you very much for your comments and thorough observations. We reflected on the points you suggested and tried to ameliorate the paper accordingly. Here are our point-by-point responses. In order to better the language quality of the paper as you suggested, we also conducted an extensive grammar check.

1. The first sentence of the abstract is hard to follow.

We agree the first sentence was probably too long and did not convey the intended meaning. We split it into two shorter periods in the revised version of the manuscript ("The association between sleep problems and quality of life has been well documented and the COVID-19 pandemic seemingly had an impact on both sleep quality and health-related quality of life (HRQoL). However, recent evidence about this relationship among university students is limited.").

2. Line 30, obstructive sleep apnea is more commonly used.

Thank you for noticing. We used obstructive sleep apnea instead.

3. Line 150-Line 152, is it a part of this manuscript?

It was indeed part of the journal template we retained by mistake. We removed the two intruded lines in the revised manuscript.

4. 1119 females vs. 160 males. The gender distribution is highly imbalanced. Did you use any method to handle it in the regression analysis? Does this affect the results?

We definitely agree that the imbalance in gender distribution is worth noticing and considering. Although it was already mentioned in the Discussion section, we expanded the paragraph dealing with this issue to address its possible impacts on the observed results (limited precision with respect to male participants, possible inflated coefficients as a by-product of the likely association between gender and the considered outcomes, and low reliability of gender comparisons) as follows:

"A large majority of the participants were female (87.5%) and given the previously re-ported associations between gender and the investigated variable [51,70,71] this highly imbalanced distribution might have had an influence on the results of the analyses conducted. Indeed, the high prevalence of female students in our sample might have inflated the regression coefficients and our results might be not relevant with respect to male students who were scarcely represented in this study. Accordingly, the results of the between-gender comparisons conducted should be considered with caution."

However, the only statistical precaution we used was inserting gender in the first regression block in order to "control" its impact on the association between sleep variables and HRQoL. We recognize that the coefficient for gender might be not reliable because of the asymmetry in gender distribution in our sample, but we chose not to use artificial corrections (weighting or similar) given the explorative purposes of our study. Our sample was indeed not representative and the discussion of this limitation was also expanded in the revised manuscript.

5. You mentioned the COVID-19 pandemic. Is poor sleep quality related to the COVID-19 pandemic in this study?

We think you spotted a relevant point. We did not considered relevant pandemic-related factors which could have had an impact on the investigated outcomes and thus, although we mentioned the importance of considering the pandemic context, we were not able to analyse its relevance in the data we collected. We added a phrase in the Discussion section to highlight this limitation ("Besides, no information was collected about contextual and individual factors related to the COVID-19 pandemic (e.g., exposure to contagion/infection, fear of contagion, feel-ings of loneliness and isolation), and thus precise inferences about the possible impact of the subjective experience of the pandemic situation on the reported findings cannot be made").

6. What method did you use to verify the reliability of questionnaires responses?

We reported Cronbach's alpha computed in our sample as an estimate of questionnaires' reliability for all the scales we used, if you were referring to this psychometric aspect. However, if you meant to address the reliability of the responsed in term of "quality" control (individual data quality, response rate, etc.) we indeed could not verify it  thoroughly. After data cleaning (duplicates removal), the responses were all "valid" and mean scores were in the acceptable range for all participants. Nonetheless, given the recruitment procedure (university social network groups, students mailing lists, students attending specific courses, etc.) it was not possible to estimate the reached sample, and considering the survey topics, self-selection and other biases cannot be excluded. We further highlighted these points in the Discussion section of the revised manuscript ("Moreover, the online recruitment procedure made it impossible to control the quality of the responses and a reliable response rate could not be obtained. Thus, selection bias could have occurred and sample self-selection should indeed be considered as a possible competitive explanation for the high rates of sleep problems and distress observed. In fact, since participants chose to take part to the study after receiving information about its aims and objectives, it is possible that students more sensitive to the investigated domains (i.e., those suffering of sleep problems) were more inclined to participate and such a selection process might have produced over-estimated prevalence rates for both poor sleep quality and insomnia. For all these reasons, the generalizability of the presented findings could be limited and comparison with previous results should be conducted cautiously.").

7. What are your eligibility criteria for this study?

The intended sample was briefly (maybe too briefly) described in the Methods section (students enrolled at Sapienza University of Rome attending undergraduate or graduate courses). To address the young-adults population, older students (>35 years) were excluded and no further exclusion criteria were applied in our study. The phrase "No further exclusion criteria were applied" was added in the Methods section to better clarify this point.

Reviewer 2 Report

Thank you for the opportunity to review this study.

I have several comments, and hope these are helpful.

  1. title: If you are looking for "health-related quality of life," it will be easier to understand if you use this term consistently in your title and paper.
  2. introduction:  For health-related quality of life section, various variables like recent pain, depression, anxiety, sleeplessness, vitality, and the cause, duration, and severity of a current activity limitation an individual may have in his or her life, was used, and I am wondering why and how you chose variables of this study.  
  3. method: It needs to report how you calculate the sample size for this study. The original or translated version of instruments'  validity or reliability needs to be reported. And I am wondering why the SF-12 was used to measure health-related quality of life. The reason why you collect data about insomnia and stress was not stated in the introduction or method section.   what cohen's d did you use and what is the evidence of this effect size and alpha?
  4. result: It is described in detail in the table, so only a few important parts need to be described, and it is necessary to describe it without a few statistical figures.  It is necessary to properly modify the table to make it easier for readers to understand the results of hierarchical regression analysis (Table 4 and 5).
  5. Discussion: Each paragraph need to be constructed as a comparative analysis with previous studies according to the main results, and it is necessary to focus on the meaning of the regression results. This content about CBT-I was not directly related to this study's findings.

Thank you.

Author Response

Thank you very much for your review. We appreciated your keen comments and tried to ameliorate the manuscript according to your suggestions.

Here are our detailed responses to the observations you made reported point-by-point.

1. title: If you are looking for "health-related quality of life," it will be easier to understand if you use this term consistently in your title and paper.

We definitely agree that using different labels across the paragraphs may be confusing. We probably exceded in it for stylistic variety and thus tried to make use of the term health-related quality of life (and the related abbreviation HRQoL) more consistently in the revised manuscript. It is indeed one of the keywords, and your comment is certainly appropriate. We also considered changing the title substituting "mental and physical health-related quality of life" with just "health-related quality of life", but decided to maintain it as it was in order to convey the dimensions of HRQoL investigated in our study (which are, as you noticed, not exhaustive of this complex construct).

2. introduction:  For health-related quality of life section, various variables like recent pain, depression, anxiety, sleeplessness, vitality, and the cause, duration, and severity of a current activity limitation an individual may have in his or her life, was used, and I am wondering why and how you chose variables of this study. 

After reading your comment we realized that although it appeared reasonable to us to explore HRQoL in relation to sleep quality, we did not make explicit our rationale for doing this (which was indeed the role of sleep in health and its well-documented associations with relevant aspects of mental and physical health). We added a paragraph in the Introduction to clarify this point in the revised manuscript (lines 46-52: "Quality of life is a broad concept that refers to the subjective perception of one’s well-being and position in life [23], and health-related quality of life (HRQoL) pertains to the health aspects of quality of life which directly reflect on physical and mental well-being. Given the above-mentioned role of sleep in health and its relations with both mental and physical conditions whose relevance for well-being is evident, it is not surprising that the association between sleep quality and quality of life has been widely investigated both in clinical [24–26] and community samples [27–30].").

3. method: It needs to report how you calculate the sample size for this study. The original or translated version of instruments'  validity or reliability needs to be reported. And I am wondering why the SF-12 was used to measure health-related quality of life. The reason why you collect data about insomnia and stress was not stated in the introduction or method section.   what cohen's d did you use and what is the evidence of this effect size and alpha?

In fact, sample size was not computed a priori for this study. Given the descriptive and substantially explorative purposes of our research we did not formulate a priori hypotheses about the relations between the investigated variables and thus we did not judged conducting a power analysis as a methodological need. For the same reasons, we chose not to conduct an a posteriori power analysis after the results were obtained. However, future cohort studies or longitudinal studies investigating specific hypotheses on the same variables will definitely require identifying an adequate sample.

With respect to the psychometric properties of the employed instruments, we reported them in a detailed manner as you suggested in the Methods section and we find that the updated version of the manuscript is more neat and clear.

We chose to use the SF-12 because of its wide diffusion in Italy and abroad and because of its well-researched validity which allows reasonable comparisons with previous results. We recognize that it is not the best instrument to measure HRQoL (the dimensions investigated by the EuroQoL family instruments are probably more exhaustive), but its concurrent validity is definitely acceptable (see for example Johnson & Coons, 1998) and so we employed it as a feasible solution.

Given the above-mentioned explorative purpose of our study, insomnia was investigated to obtain a rough estimate of the prevalence of the symptoms of this very common sleep disorder in our sample (furthermore, the ISI is probably the most used instrument in studies investigating sleep problems in both clinical and non-clinical populations). Perceived stress was considered mainly to control for the role of contextual psychological distress in the regression models, but its value as an informative psychological variable has been widely reported. We included these considerations in the revised manuscript, both in the extended description of the instruments and in the Discussion section.

Finally, we computed Cohen's d as standardized mean difference (Cohen, 1988) according to the updated guidelines proposed by Lakens (2013) for between-subjects designs on independent samples (Lakens report Cohen's d as simply d in these cases). Cohen's conventional criteria (Cohen, 1988, in references) were used to evaluate the magnitude of the effect sizes reported (with the main purpose of comparing the magnitude of the observed differences in HRQoL among poor vs good sleepers and subjects with insomnia vs subjects without insomnia). Cronbach's alpha was computed with the conventional formula considering the sum of the items' variances and the total score variance (routinely implemented in SPSS). Adequacy criteria suggested by Nunnally and Bernstein (1994, in references) in their handbook were considered.

4. result: It is described in detail in the table, so only a few important parts need to be described, and it is necessary to describe it without a few statistical figures.  It is necessary to properly modify the table to make it easier for readers to understand the results of hierarchical regression analysis (Table 4 and 5).

Thank you for noticing. Tables were indeed very difficult to read and their format had not been adjusted for the journal template in the manuscript version you received. We made them more readable according to your suggestion and we find that the Results section is clearer and easier to read in the revised manuscript.

5. Discussion: Each paragraph need to be constructed as a comparative analysis with previous studies according to the main results, and it is necessary to focus on the meaning of the regression results. This content about CBT-I was not directly related to this study's findings.

We agree with your observation. The Discussion dealt too briefly with relevant results and the section focusing on the regression models missed many important points, including a comparative analysis of the results. We expanded that part with references to the results of two previous studies which employed a research design similar to that of our work and measured the same constructs commenting on similarities and discrepancies (lines 306-327 in the revised manuscript: "These findings are substantially consistent with those reported in previous studies that considered the same variables and used the same measure of HRQoL (namely, the SF-12), although some differences should be considered. In particular, Darchia and colleagues [24] found significant associations between sleep quality (PSQI global score) and both physical and mental HRQoL after controlling for socio-demographic variables, BMI, and depression in a sample of 395 subjects aged between 20 and 60 years from Georgia, but in their study the magnitude of the unique association between sleep quality and physical HRQoL was less consistent than that we observed in our sample. However, this discrepancy might be attributable to the wider age range of the subjects included in the research of Darchia et al., since it is likely that several other uncontrolled factors beyond sleep quality had an impact on the older subjects’ perception of their physical health. On the other hand, in the study conducted by Clement-Carbonell and colleagues [28], the same pattern of associations was observed among 337 university students from Spain after controlling for stress and health habits (diet and exercise). Consistently with our findings, significant relations were found between physical HRQoL and both the subjective sleep quality and the sleep disturbances components of the PSQI, but in comparison with the result of our study the association observed between sleep quality components and mental HRQoL is stronger. This disparity might be due to the above-mentioned variance overlap between the PSS-10 and the SF-12 MCS and to the overall high level of perceived stress and low mental HRQoL we found in our sample, which may have deflated regression coefficients in the model predicting the MCS.").

We agree that a review of effective intervention was out of the scope of the paper, but we find that a brief mention of the possible implications of our findings for interventions development could be useful. Thus, according to the suggestions of other reviewers, we chose to maintain the paragraph reporting previous results about CBT-I exploring in detail the relevance of CBT-I strategies with respect to the outcomes investigated in our study and its possible integration in broad range interventions for students' well-being.

Reviewer 3 Report

"Sleep quality and its associations with physical and mental 2 health-related quality of life among university students: a 3 cross-sectional study" can be published after revision.

  • format needs improvements
  • language should be improved
  • design could be clearer
  • conclusion should be extended

Author Response

Thank you very much for your comments and observations.

With respect to the specific points you mentioned, we made several changes in the revised version of the manuscript. In particular, we improved the visual format of the paper and made the tables easier to read by re-ordering the rows labels and separating the two steps of the regression models.

We also conduct an extensive grammar check to ameliorate the quality of the English language and pointed out the research design in the Methods section specifying inclusion criteria and the properties of the employed instruments.

Major changes were made in the Discussion section, where the results of the regression models were examined in depth with respect to previous findings. Besides, we better explored the limitations of the study and clarify the implication of the results for the development of interventions.

Reviewer 4 Report

For this cross-sectional study, the authors primarily investigated the association between sleep quality and physical and mental health-related quality of life (HRQoL) in university students attending a large Italian university.

The findings suggest that poor sleep quality negatively affects the HRQoL. Authors further report that when sociodemographic and stress variables are statistically controlled for, sleep quality has a greater effect on the physical HRQoL as compared to its effect on mental HRQoL. Authors then cite CBT-I strategies to promote better sleep for university students.

Although recruitment was limited to only one university, majority of the results presented are largely consistent with those reported in the literature. Few major limitations of the study – not collecting data regarding other sleep disorders and use of self-luminous devices, are appropriately acknowledged.

Overall, the authors present the results clearly and the manuscript is well drafted.

Minor comments:

Mention participant demographics in the abstract

Page 3, Line 129 – Incorrect abbreviation. Do authors mean MCS?

Page 4, Lines 150-152 – This text should be removed

Page 4, Lines 158-168 – Confidence interval can be abbreviated as “CI” for easier read

Page 7, Line 218 – was found “for?” physical exercise?

Page 7, Line 230-231 – delta R2 seems to be reported for the Adjusted R2? Also, it says the contribution of the sleep quality block was significant for MCS. However, the F statistic was lower for Step 2 in Table 5.

Page 8, Line 234 – Only daily disturbances were reported to influence both PCS and MCS HRQoL. Authors could discuss the implications of this finding further. Does CBT-I lower daily disturbances?

Author Response

Thank you very much for your comments and accurate observations. We really appreciated your suggestions and tried to ameliorate the manuscript accordingly as you can see in the uploaded revised version.

Here are our detailed responses to your comments (your comments are reported in bold):

- Mention participant demographics in the abstract.

Participants demographics were added in the abstract.

- Page 3, Line 129 – Incorrect abbreviation. Do authors mean MCS?

- Page 4, Lines 150-152 – This text should be removed

- Page 4, Lines 158-168 – Confidence interval can be abbreviated as “CI” for easier read

- Page 7, Line 218 – was found “for?” physical exercise?

Thank you for noticing. It was indeed MCS, it was corrected in the revised manuscript. The template example text was erased (lines 150-152).

The abbreviation CI was used as you suggested in the revised manuscript and the misprint was corrected.

- Page 7, Line 230-231 – delta R2 seems to be reported for the Adjusted R2? Also, it says the contribution of the sleep quality block was significant for MCS. However, the F statistic was lower for Step 2 in Table 5.

In fact, the value reported as delta R-squared is correct.

It is true that the F statistic was lower for Step 2 in the regression model predicting MCS. It is due to the fact that F depends on the number of predictors added in each step (with related changes in degrees of freedoms in numerator and denominator).

However, the F value tests the overall significance of the model and does not provide useful information for model comparison. The sleep quality block contribution was inferred by the significance of the R-squared change F statistic. Tables have been adjusted in the revised version to make the visualization of the results easier.

- Page 8, Line 234 – Only daily disturbances were reported to influence both PCS and MCS HRQoL. Authors could discuss the implications of this finding further. Does CBT-I lower daily disturbances?

We revised and extended the Discussion section according to your suggestions. We added a recent meta-analysis in the references dealing with the impact of CBT-I on sleep quality as measured by the PSQI and daily disturbances (van Straten et al., 2018, number [66] in the revised manuscript) and made explicit the link between this treatment solution and the findings of our study.

Round 2

Reviewer 1 Report

The paper is considerably improved from the previous version.

Author Response

Thank you again for your comments and your quick reply.